# Uncertainty quantification and sensitivity analysis of COVID-19 exit strategies in an individual-based transmission model

**Federica Gugole**[1], **Luc E. Coffeng**[2], **Wouter Edeling**[1], **Benjamin Sanderse**[1], **Sake J. de Vlas**[2], **Daan Crommelin**[1,3]*

**1** Centrum Wiskunde & Informatica, Amsterdam, The Netherlands, **2** Department of Public Health, Erasmus MC, University Medical Center Rotterdam, Rotterdam, The Netherlands, **3** Korteweg-de Vries Institute for Mathematics, University of Amsterdam, Amsterdam, The Netherlands

* daan.crommelin@cwi.nl

**Data Availability Statement:** The computational model considered here is publicly available on GitLab https://gitlab.com/luccoffeng/virsim/-/tree/

## Abstract

Many countries are currently dealing with the COVID-19 epidemic and are searching for an exit strategy such that life in society can return to normal. To support this search, computational models are used to predict the spread of the virus and to assess the efficacy of policy measures before actual implementation. The model output has to be interpreted carefully though, as computational models are subject to uncertainties. These can stem from, e.g., limited knowledge about input parameters values or from the intrinsic stochastic nature of some computational models. They lead to uncertainties in the model predictions, raising the question what distribution of values the model produces for key indicators of the severity of the epidemic. Here we show how to tackle this question using techniques for uncertainty quantification and sensitivity analysis. We assess the uncertainties and sensitivities of four exit strategies implemented in an agent-based transmission model with geographical stratification. The exit strategies are termed Flattening the Curve, Contact Tracing, Intermittent Lockdown and Phased Opening. We consider two key indicators of the ability of exit strategies to avoid catastrophic health care overload: the maximum number of prevalent cases in intensive care (IC), and the total number of IC patient-days in excess of IC bed capacity. Our results show that uncertainties not directly related to the exit strategies are secondary, although they should still be considered in comprehensive analysis intended to inform policy makers. The sensitivity analysis discloses the crucial role of the intervention uptake by the population and of the capability to trace infected individuals. Finally, we explore the existence of a safe operating space. For Intermittent Lockdown we find only a small region in the model parameter space where the key indicators of the model stay within safe bounds, whereas this region is larger for the other exit strategies.

## Author summary

Many countries are currently dealing with the COVID-19 epidemic and are looking for an exit strategy such that life in society can return to normal. For that purpose computational

v1.0.5. The scripts used for the generation of the UQ and SA campaigns and post-processing of the results are publicly available on GitHub https://github.com/FGugole/UQ_covid19. The data used for the analysis have been generated by the aforementioned scripts.

**Funding:** FG, WE and DC were supported by the European Union Horizon 2020 research and innovation program (https://ec.europa.eu/programmes/horizon2020/en) under grant agreement 800925. LEC acknowledges funding from the Dutch Research Council (NWO https://www.nwo.nl/en), grant number 016.Veni.178.023. LEC, SJdV and DC received support from ZonMw (https://www.zonmw.nl/en/) grant number 10430022010001. The funders had no role in study design, data collection and analysis, decision to publish, or preparation of the manuscript.

**Competing interests:** The authors have declared that no competing interests exist.

models are used to predict the spread of the virus and to assess the efficacy of policy measures before putting them into practice. These models are subject to uncertainties (due to, for instance, limited knowledge of the parameter values), which can lead to a large variability in model predictions. It is therefore fundamental to assess which range of values a model produces for key indicators of the severity of the epidemic. We present here the results of the uncertainty and sensitivity analysis of four exit strategies simulated with an individual-based model of the COVID-19 transmission. As key indicators of the severity of the pandemic we consider the maximum number of cases in intensive care and the total number of intensive care patient-days in excess. Our results show the crucial role of the intervention uptake by the population, of the reduction in the level of transmission by intervention and of the capability to trace infected individuals.

This is a *PLOS Computational Biology* Methods paper.

## Introduction

Many countries are currently dealing with the COVID-19 epidemic and are searching for an exit strategy such that life in society can return to normal. However, in absence of an effective curative treatment and, until recently, of an effective vaccine, non-pharmaceutical interventions have been used to keep case numbers as low as possible. In the past there have been numerous other epidemics during which government actions were required to protect the population. Examples are the influenza pandemic (also known as Spanish flu) in 1918 or the more recent Mexican flu (or swine flu) in 2009 [1]. When an infectious disease outbreak occurs, governments rely on computational models to predict the spread of the disease and to explore the potential impact of interventions [2–4]. Thus, computational models enable decision makers in government and public health institutions to assess the efficacy of policy measures before actual implementation.

Since computational modeling of the epidemic has come to play a significant role for informing policy, it is important to assess the uncertainties of the models and of their predictions. Such uncertainties can stem, for instance, from limited knowledge about the values of the input parameters or from the intrinsic stochastic nature of part of the computational models. The presence of any of these uncertainties leads to uncertainties in the model predictions. A central question is therefore what distribution of values is produced by the model for key indicators of the severity of the epidemic and of the intensity of interventions.

In this study we present results from an analysis of uncertainties and sensitivities of an agent-based model of the COVID-19 epidemic [5, 6], obtained using techniques of Uncertainty Quantification (UQ) and Sensitivity Analysis (SA) [7, 8]. UQ is an area of mathematics dealing with propagation of uncertainties from model input to model output, and establishing model input uncertainties from observation data. Let us denote with $\mathcal{M}$ the computational model and let $X$ and $Y$ be its input parameters and output quantities, respectively, such that $Y = \mathcal{M}(X)$. $X$ has probability distribution $p(X)$, reflecting its uncertainty. A central aim of UQ is to determine the probability distribution of $Y$, i.e. $p(Y)$, given $p(X)$. With SA, one determines what fraction of the uncertainty of $Y$ (e.g. fraction of the total variance of $Y$) is due to individual elements of the vector $X$, given $p(X)$. Thus, one searches which input parameters generate the most (or the least) variance in $Y$. It is frequently the case that a large part of the variance of an output quantity is due to a subset of the input parameters involved.

The UQ and SA frameworks employ a probability distribution—as opposed to a single value—to describe each input parameter. These distributions are typically determined from available data or from expert knowledge. With proper probability distributions assigned to each input parameter, the UQ and SA frameworks can be used to assess whether exit strategies are robust. A useful criterion for robustness is that, given the model input distributions, the 95[th] percentile of a chosen output quantity—usually termed Quantity of Interest (QoI)—remains below a critical threshold. Examples of QoI in the context of epidemic modeling are the peak number of COVID-19 patients in intensive care (IC) units and the total number of fatalities due to COVID-19. For other QoIs, e.g. life years gained, a more relevant criterion is that the 5[th] percentile stays above a minimum value. This can be dealt with in an analogous manner.

The aim of this work is to perform a model-based quantitative analysis of uncertainties and sensitivities for the computational representations of four exit strategies, and assess the uncertainties in model simulation results. We show how such analysis can be performed by means of computational methods and concepts from the fields of uncertainty quantification and sensitivity analysis, and what kind of insights can be obtained. UQ is important when decision makers have to choose between interventions, guided by model-based predictions of what the effects of these interventions will be. Consider a situation where, looking at a single prediction based on the best point estimate for the model input parameters, intervention A appears more attractive than intervention B (e.g. because the predicted peak number of COVID-19 patients in IC is lower under A). However, when taking parameter uncertainties into account, UQ may show that intervention A also gives a higher probability of disastrous outcomes than B (e.g. higher chance of peak numbers that overwhelm IC capacity). This latter insight can be reason to choose B over A. SA provides a different kind of insight: it shows which parameters have the most effect on the outcome and on its uncertainty, and thereby which parameters are the most important ones to control or influence through policy. For example, in case of the Contact Tracing strategy considered in this paper, is it more effective to spend additional resources on tracing *more* infected individuals, or on tracing them *faster* (i.e., reduce the delay between becoming infected and being identified as such), in order to reduce the peak number of IC patients (or the variance in this peak number)? SA can help to answer such questions.

In order to perform our analysis we consider the spread of the COVID-19 disease in the Netherlands in the context of an open-source agent-based model with geographical stratification [5, 6]. We note that the framework of our analysis is not restricted to this specific model, but can also be applied to other strategies, to more complex models and to other epidemics than COVID-19.

## Methods and model

In what follows, we provide an overview of the computational model used in our analysis and the exit strategies implemented in the model. Next, we summarize some key concepts of uncertainty quantification and describe different sources of uncertainties. We discuss also the chosen SA method and the quantities of interest selected for our study. We conclude this section with a description of the computational UQ and SA framework that we employ.

### Computational model

We employ the publicly available virsim model [6], which is a stylized representation of the COVID-19 epidemic with geographical stratification of both transmission and interventions (i.e., a meta-population model). More specifically, it is an agent-based model with a geographical structure defined by means of clusters (representing for instance towns and villages) and

**Table 1. Model parameters varied in the UQ analysis.** Values represent the mean and 95%-confidence interval (95%-CI) of the input distributions.

| Input parameter | Strategy | Distribution | Mean | 95%-CI |
|---|---|---|---|---|
| intervention_effect | FC | Beta($\alpha = 38, \beta = 70$) | 0.35 | [0.27, 0.44] |
| uptake | FC, PO, IL | Beta($\alpha = 16, \beta = 2$) | 0.89 | [0.71, 0.99] |
| trace_rate_I | CT | $\Gamma$(shape = 2, scale = 0.2) | 0.40 | [0.05, 1.11] |
| trace_prob_E | CT | Beta($\alpha = 2, \beta = 6$) | 0.25 | [0.04, 0.58] |
| trace_contact_reduction | CT | Beta($\alpha = 10, \beta = 2$) | 0.83 | [0.59, 0.98] |
| lock_length | IL | $\Gamma$(shape = 20, scale = 2) | 40.0 | [24.4, 59.3] |
| lift_length | IL | $\Gamma$(shape = 15, scale = 1) | 15.0 | [8.4, 23.5] |
| lock_effect | IL | Beta($\alpha = 14, \beta = 42$) | 0.25 | [0.15, 0.37] |
| intervention_lift_interval | PO | $\Gamma$(shape = 25, scale = 2) | 50.0 | [32.4, 71.4] |
| pl_intervention_effect_hi | PO | Beta($\alpha = 14, \beta = 42$) | 0.25 | [0.15, 0.37] |
| avg_duration_infectiousness | non-policy-related | $\Gamma$(shape = 25, scale = 0.2) | 5.00 | [3.24, 7.14] |
| $R_0$ | non-policy-related | $\Gamma$(shape = 100, scale = 0.025) | 2.50 | [2.03, 3.01] |
| intervention_effect_var$^{-1}$ | non-policy-related | $\Gamma$(shape = 2, scale = 0.05) | 0.10 | [0.01, 0.28] |
| shape_exposed_time | non-policy-related | $\Gamma$(shape = 17.5, scale = 1) | 17.5 | [10.3, 26.6] |

superclusters (e.g. provinces or other administrative units for which policy decisions are made). Individuals are separated into susceptible (S), exposed (E), infectious (I) and recovered (R), and life-long immunity is assumed. The addition of geographical stratification allows for heterogeneity at the individual and sub-population level, as well as the implementation and evaluation of regional approaches. A technical description of the model is provided in S1 Text and S1 Table. For further details on the model we refer the reader to de Vlas & Coffeng [5]. For this study, we employed an updated quantification of the model [9]; see Table 1 in the main text (with the corresponding distributions displayed in S1 Fig) and S1 Table for an overview of the parameter values.

We consider four exit strategies: Flattening the Curve (FC), Contact Tracing (CT), Intermittent Lockdown (IL) and Phased Opening (PO). Each strategy is part of the model implementation (see S2 Text for the computational details) and is defined by a unique set of parameters. More details about the strategies parameters are given in a later section, where we discuss also the input distributions that have been considered to account for the respective uncertainties.

Since this study is of conceptual nature and does not aim to model real-world scenarios as accurately as possible, we simulate a population of 1 million individuals and we focus on the first year after the implementation of a strategy (the idea being that governments will regularly evaluate and adapt the strategy in use). In the Discussion section we consider what is needed to bring our analysis to more complex models (and strategy implementations) that aim to be more realistic. All simulated intervention strategies are preceded by a period of lockdown, analogous to the situation in the Netherlands from March to May 2020. During this period, we assume that an intervention package is implemented that heavily reduces transmission. The model parameter for this effect (intervention_effect) is expressed in terms of the level of transmission that the intervention package still allows, relative to a situation without any interventions (i.e., 100% minus the reduction in the overall transmission rate). Based on a previous quantification of the model using extensive data on the Dutch COVID-19 outbreak [9], we assume that the lockdown reduces transmission to 30% on average for one week and then to 15% on average after the Dutch government introduced stricter measures. In accordance with the Dutch situation, the effect of the lockdown is simulated for 60 days, followed by a period of partial lockdown (assumption: intervention_effect = 25%), analogous to

the re-opening of schools in the Netherlands [9]. After another 30 days, one of the exit strategies takes effect in the simulation. Overall we simulate a period of 550 days corresponding to roughly 1.5 years; see S2 Text for details about how this is implemented in the model.

## Uncertainty and its quantification

Uncertainties in model output can arise from different sources; we discuss here four main types. The first is parameter uncertainty, referring to uncertainties in model parameters whose values can be set directly by the model user via the inputs of the computational model. An example is the reduction of the transmission rate due to the introduction of an intervention. For the COVID-19 epidemic, a significant decrease in the transmission rate following the adoption of such measures has been recorded [10–13], however the magnitude of this reduction is uncertain.

The second type of uncertainty, which we call intrinsic uncertainty, arises when a computational model is inherently stochastic. In epidemiology, many models are agent-based and possess internal stochasticity, for instance in the randomized interactions between agents. Model users often have little control over such internal stochasticity as they can typically only set the seed of the random number generator at the start of a simulation [14].

The third type is model-form uncertainty, referring to uncertainty or errors in the structure of the model itself (e.g. due to transmission mechanisms not represented in the model). This type of uncertainty cannot be analyzed by changes in the model inputs but requires a comparison either with independent observation data or with other models—as done for example in climate science [15].

Lastly, initial condition uncertainty is due to the inaccuracies in the specification of the initial state of the model (i.e., the state of the simulated population at the start of the model run). Since we consider here model outputs which are independent from the specific timing of e.g. epidemic peaks, this type of uncertainty is not important in our analysis.

In this study we analyze the parameter and intrinsic uncertainties by means of non-intrusive UQ methods, which means that we treat the computational model (i.e., the virsim model) as a black box. We extend the notation from the introduction to $Y = \mathcal{M}(X, r)$, where $r$ denotes the seed of the random number generator. By random sampling of the parameter vector $X$ from its distribution $p(X)$ and altering the seed $r$, followed by executing the model $\mathcal{M}$, we create random samples from the probability distribution of $Y$. Thus, we can probe the uncertainty of $Y$ and estimate its distribution by repeated model executions.

## Sensitivity analysis

In order to assess which parameters create the most uncertainty in the model output $Y$ under the different strategies, we compute the Sobol indices [16]. This is a form of global sensitivity analysis focusing on the variance of $Y$. Assuming mutual independence among the input parameters, $\mathrm{Var}(Y)$ is decomposed into fractions which can be attributed either to a single input parameter (first order Sobol indices) or to a set of parameters (higher order Sobol indices). Given $p(X)$, the first order Sobol index for the $i$-th input parameter ($X_i$) is defined as $S_i \coloneqq \mathrm{Var}_i[\mathbb{E}_{\sim i}(Y|X_i)] / \mathrm{Var}(Y)$, as explained in S3 Text. We use the notation $\mathbb{E}_i$ for expectation over $X_i$ and $\mathbb{E}_{\sim i}$ for expectation over all $X_1, X_2, \ldots$ except $X_i$ (and similarly for $\mathrm{Var}_i$ and $\mathrm{Var}_{\sim i}$). If $S_i$ is close to 1, it means that the variance of $Y$ is almost entirely due to the variance of $X_i$. The overall effect on the model output of all parameter combinations involving $X_i$ is given by the total Sobol index $S_{T_i} \coloneqq 1 - \mathrm{Var}_{\sim i}[\mathbb{E}_i(Y|X_{\sim i})] / \mathrm{Var}(Y)$. We remark that the Sobol index for the seed of the pseudo-random number generator can be non-negligible. It quantifies the contribution of the internal stochasticity to the overall variance of the output, i.e. the QoIs.

Regarding the assumed mutual independence of the input parameters, there might be dependencies when parameters are actually estimated from data. However such dependencies are not implemented in the computational model, hence the selected input parameters (see below for more details) are effectively treated as mutually independent. We refer to S3 Text for more details about the theory of the Sobol indices.

## Quantities of interest

In our analysis we consider the model predictions for the number of incident and prevalent individuals in the population that require IC admission. As IC capacity is limited, the question whether the IC capacity will be exceeded (and if so, by how much) according to model predictions is clearly important. To investigate this, for each simulation we consider two quantities of interest (QoI):

1. the maximum of the moving average of the prevalent cases in intensive care (averaging window = 30 days)

2. the total number of IC patient-days in excess of IC bed capacity (referred to as "IC patient-days in excess" from hereon).

The first QoI shows the peak value of the number of COVID-19 patients in IC units, giving an indication of the intensity of an outbreak. We apply a moving average to focus on longer-term trends, filtering out short-term "noisy" variations. When analyzing robustness, a natural threshold for this QoI is the available IC capacity, which may vary from country to country and from month to month. In our analysis we assume the IC capacity to be constant in time and we consider the Netherlands as reference country. De Vlas and Coffeng [5] calculate that during the first epidemic wave in 2020 the Dutch maximum IC capacity for COVID-19 patients was 109 IC beds per million inhabitants.

The second QoI quantifies by how much the total IC capacity is overburdened. It is defined as $\sum_{t=0}^{T} \max(0 , p(t) - p^*)$ where $p(t)$ is the number of IC prevalent cases at day $t \in \{0, 1, 2, \ldots, T\}$, $p^*$ is the maximum IC capacity and the summation runs from the start time of the simulation ($t = 0$ day) to the final time ($t = T$ days). It shows the extent to which the government may have to rely on other countries, e.g. Germany, to take in IC patients from the Netherlands. Setting a threshold for this QoI is complicated, as many economical and political factors come into play here.

We note that the two QoIs as defined here do not depend on time. The first QoI is defined as a maximum over the time interval of simulation, whereas the second QoI is defined as a summation over the same time interval. Thus, a single model simulation yields a single, time-independent value for each QoI.

## Policy-related uncertainties

The implementation of a strategy in the computational model is determined by a set of input parameters for the model. Some of the parameters can be controlled (up to a certain degree) by policy makers. These are parameters related to social aspects of the population—e.g. social distancing—and to the availability of resources, e.g. to test and track infected individuals. They can be controlled to some extent by government authorities through imposing stricter (or less strict) rules for e.g. social distancing, or through making more (or less) resources available. We call them policy-related parameters and we refer to the uncertainties generated by these parameters as policy-related uncertainties.

Below we provide details about which policy-related parameters are treated as uncertain, together with the rationale behind the probability distributions chosen for them. The distributions, together with their mean and 95%-confidence intervals are provided in Table 1; their plot instead is provided in S1 Fig.

We use Beta distributions for those parameters that are naturally restricted to values on a finite interval, such as those representing probabilities or percentages. We remark that this allows us to use distributions that are very different from the uniform distribution. With the hyperparameters used here, the probability density of the Beta distribution decreases to zero towards the boundaries of the support. By contrast, the uniform distribution attributes equal probability to any value within the support of the distribution. For parameters defined as a time scale or a rate we choose the $\Gamma$ distribution as it has one unique peak (with the hyperparameters used here) and a semi-infinite domain (all non-negative values).

Our choice of the distributions was guided by expert knowledge. Ideally, as more data become available the initial guesses for those parameters that can be directly estimated could be updated to better reflect the data (although this has not been done here). However, this is not always possible as there may be parameters that cannot be estimated directly from data. Furthermore, parameters can be model specific. Relatedly, some of our selected distributions are not transferable to other transmission models if these have explicit representations (with dedicated parameters) of processes only implicitly represented in virsim. For instance, in the virsim model the parameter for the effect of the intervention in Flattening the Curve implicitly includes the effect of different measures (e.g. social distancing, working from home and closure of schools), therefore it cannot be directly estimated from data for every plausible combination of interventions. Hence it is advisable to combine expert knowledge and calibration on the available data to determine the most appropriate input distributions.

**Characteristics of the four exit strategies.  Flattening the Curve**. The main idea of FC is to gradually resume activities that have been interrupted by the lockdown such that the virus spreads among the population at a lower pace while keeping the pressure on the healthcare system manageable. Here we model only the first intervention after the lockdown with the idea that the evolution of the pandemic can be appropriately reconstructed up to mid of June 2020, such that the focus is on the next step to take. In the computational model, this strategy is governed by two parameters:

- the effect of the intervention (model parameter `intervention_effect`). The effect of the intervention package is expressed in terms of the level of transmission it still allows, relative to a situation without any intervention (i.e., one minus the reduction in the overall transmission rate). This parameter is supposed to allow a slight increase in the average contact rate with respect to the situation of general lockdown with schools open. We sample it from a Beta distribution;

- the uptake of the intervention by the population, model parameter `uptake`, which we draw from a Beta distribution. We assume that the majority of the population would adhere to the introduced measures, hence the bulk of the distribution is moved towards values close to 1.

**Contact Tracing**. This strategy consists of tracking potentially infectious contacts such that only infected people drastically reduce their interactions with others, thus limiting the spreading of the virus while allowing events and activities to take place nonetheless. Three factors characterize such an approach:

- the delay between becoming infectious and being identified as such (if at all); represented by the inverse of the model parameter `trace_rate_I`. A fast identification is technologically

very challenging, hence we give higher probabilities to identifications requiring more than 20 hours. We employ a Γ distribution for the inverse of the delay;

- the probability of an infected contact to be identified before the person turns infectious; represented by the model parameter `trace_prob_E`. Such a classification is challenging as tracking data might be incomplete or there might not be enough capacity to process them. Therefore we choose a Beta distribution with relatively large variance and mean shifted towards lower values;

- the quality of the isolation and its effect on transmission; represented by the model parameter `trace_contact_reduction`. We assume that whoever is being tracked would adopt every possible measure to avoid transmitting the virus further. For this parameter we use a Beta distribution.

**Intermittent Lockdown**. The idea here is to alternate periods of lockdown and periods of complete opening. Ideally the start of these periods would be determined by, e.g., a number of COVID-19 IC patients above or below a certain threshold. However such a dynamic trigger for the start of the lockdown/opening is not included in the current version of the virsim model. Therefore we alternate periods of lockdown and periods of opening at preset intervals. When such a strategy is to be modeled, the following parameters have to be chosen:

- the duration of the lockdown and of the following lift of measures represented by the model parameters `lock_length` and `lift_length`. Since a long lockdown might have a heavy impact on the psychological health of the population with more people, for instance, becoming depressed, we try to keep a balance between the lockdown and the lift period. We consider Γ distributions both for the lockdown and for the lift periods;

- the effect of the lockdown (model parameter `lock_effect`) expressed in terms of the level of transmission it still allows, relative to a situation without any intervention. We assume it to have (on average) an effect in contact reduction similar to a general lockdown with schools open. We sample it from a Beta distribution;

- the `uptake` of the intervention by the population. For this parameter we use the same Beta distribution assumed in case of FC.

**Phased Opening**. The approach of the Phased Opening strategy is to release the general lockdown at the regional level and distribute the patients in need of hospital care at the national level. In this way the spread of the virus is limited to a smaller portion of the population and the burden on the healthcare system is reduced [5]. In our simulations we consider a number of phases equal to the number of superclusters, so at each phase only one supercluster lifts the lockdown. Other important features defining the strategy are:

- the time interval between one phase and the next, determined by `intervention_lift_interval`. A too short period would lead the strategy to be rather inefficient and to a strong overburden on the healthcare system in a short time. On the other hand a too long period would lead to a rather long waiting for some regions and hence to growing discontent among the population. We try to find a compromise in the choice of the Γ distribution for this parameter;

- the effect of the measures in the areas where the lift has not been applied yet, represented by the model parameter `pl_intervention_effect_hi` and expressed in terms of the level of transmission it still allows, relative to a situation without any intervention. In these areas we assume that the reduction in the average contact rate is analogous to the reduction

obtained during the periods of lockdown in case of the IL strategy, hence we use the same Beta distribution;

- the uptake of the intervention by the population, model parameter `uptake`, which is sampled from a Beta distribution, as in the cases of FC and IL.

### Other uncertainties

Besides the parameters that can be influenced by policy, discussed in the previous section, the computational model has other parameters that we include in our analysis. Examples are the reproduction number of the virus and the duration of infectiousness. Because these parameters have to be estimated from data, their values can be rather uncertain, especially when available data is scarce (for instance in the early stages of an epidemic with a new virus). For some parameters, care has to be taken when sampling them from probability distributions, since certain basic characteristics, like the doubling time, have to match the time evolution of the epidemic as captured in the (real-world) observation data. We chose distributions whose mean is (almost) the value obtained in a previous quantification of the model [9]. Using $\Gamma$ distributions was a modelling decision and the overall shape of the distributions was guided by expert knowledge.

Altogether, given the setup of the virsim model, we consider the following non-policy-related uncertain parameters:

- the average duration of infectiousness (model parameter `avg_duration_infectiousness`) has been estimated to be about 5 days. To account for errors and a conceivable lack of data in the estimation process, we sample it from a $\Gamma$ distribution;

- the reproduction number $R_0$ is determined in the model as the product between the average duration of infectiousness and the contact rate. By default it is set at $R_0 = 2.5$ and in what follows we account for uncertainties on how the virus spreads among the Dutch society. Thus we draw $R_0$ from a $\Gamma$ distribution and derive the average contact rate as the ratio between $R_0$ and the average duration of infectiousness;

- `intervention_effect_var` represents the inter-individual variation in the effect of the intervention in case of uptake (in the default setup no inter-individual variation is allowed). Such variations might be due, for instance, to the presence of children in the household. `intervention_effect_var` can assume any positive value and, for simplicity of representation, it is modeled as $1/x$ where $x$ is sampled from a $\Gamma$ distribution. Please note that even though the 95%-confidence interval for $x$ itself is narrow, the resulting interval for $1/x$ is quite large;

- in order to consider uncertainties in how the incubation period varies among the population, we sample the shape parameter of the distribution of `exposed_time` from a $\Gamma$ distribution.

The virsim model is stochastic, and as discussed in an earlier section, this internal stochasticity is a source of uncertainty of the model output. It is not possible to choose the probability distribution of the internal (or latent/hidden) random variables of the model, or to set them by hand to specific values according to some sampling plan. The only form of control as model user is to pick the seed for the pseudo-random number generator at the start of the simulation. An example of uncertainties in the model output due to the internal stochasticity is provided in S2 Fig for the FC strategy with fixed parameters but varying seed. It can be seen that the peak value of the number of prevalent cases in IC for different realizations can vary between

150 and 200 leading to differences up to 25% among the simulations. This shows that the model has significant internal stochasticity. We draw the seed from a discrete uniform distribution with support between 16384 and 65536. This amounts to independent random sampling of the internal random variables of the virsim model.

Furthermore, when using models with a geographical structure like the virsim model, uncertainties can arise from the level of geographical mixing that is allowed in the model. For sake of simplicity we do not consider such uncertainties in the present study, but they should be considered in more comprehensive studies if these are intended to inform policy makers.

We report in Table 1 the assumed input distributions, their mean and 95%-confidence intervals reflecting parameter uncertainty for model parameters that were considered in the UQ analysis. The plot of the distributions themselves is provided in S1 Fig. Finally, in S1 Table we list the model parameters that have fixed values (i.e., the parameters not considered as uncertain) in this study. The reported values have been taken from the model quantification used in [9].

## UQ and SA computational framework

If the dependence of the QoIs on the parameters is smooth and the number of uncertain parameters is not too high, the propagation of uncertainties from parameters to QoIs can be assessed with techniques such as Polynomial Chaos Expansion and Stochastic Collocation [7, 8, 17], that allow to obtain the relevant information with relatively little computational time. However, due to the presence of intrinsic uncertainty, the continuous relation between QoIs and inputs is lost as the QoIs do not depend continuously on the random seed. When this type of uncertainties is present, the aforementioned methods like Polynomial Chaos Expansion are not suitable. Therefore, we resort to Monte Carlo (MC) sampling as it is guaranteed to return reliable results (albeit with a low convergence rate). We base our UQ results on the evaluation of an MC ensemble with 2000 simulations in total. MC sampling implies that for each simulation, the full vector of uncertain parameters is sampled anew (randomly). More advanced techniques meant to overcome the issue of the internal stochasticity are currently being developed, see for instance [18] but require further investigation before application to high-dimensional systems.

For the SA, we compute the first order Sobol indices with the cost effective algorithm of Saltelli [19] (see S3 Text), and their 95%-confidence intervals are computed via bootstrapping. By performing an intelligent sampling of the input parameters, this algorithm reduces the computational cost of the calculation of the first order Sobol indices from $M^2 p$ to $M(p + 2)$ model simulations, where $M$ is the number of MC samples and $p$ is the number of uncertain parameters (including the random seed). For more details about the practical implementation of the Saltelli algorithm please see [8]. Due to the specific sampling required by the Saltelli algorithm, the SA cannot be performed on the same set of data employed for UQ. Therefore we run a separate set of simulations, and use $M = 2000$ MC samples per parameter involved in the implementation of the strategy (SA is in general computationally more demanding than UQ). This results, e.g., in case of FC in a total of $M(p + 2) = 2000(3 + 2) = 10000$ simulations, where we considered only the random seed and the policy-related parameters.

Sampling and post-processing analysis are done using the Monte Carlo sampler of the publicly available Python library EasyVVUQ [20–22]. We run the required model simulations in parallel on a supercomputer at the Poznan Supercomputing and Networking Center. For the job submission to the supercomputer we use the FabSim3 [23, 24] and QCG-PilotJob [25] packages. FabSim3, QCG-PilotJob and EasyVVUQ are all part of the open source VECMA Toolkit (VECMAtk) [26, 27]. The codes employed for this study can be found on GitHub [28].

## Results

To provide intuition for the behaviour of the model and the QoIs defined earlier, we show in Fig 1 time series of the number of prevalent cases in IC and of the number of IC patient-days in excess, for four simulations of the virsim model under the Phased Opening strategy. For each simulation, different policy-related parameters are used. It can be seen in the left panel of Fig 1 that the number of prevalent cases in IC has noisy perturbations on top of the main signal. As discussed earlier, we apply a moving average (averaging window = 30 days) to filter out these short-term noisy perturbations. The resulting smoothed time series are shown in the central panel of Fig 1. The peak value reached in each smoothed time series is our first QoI. In the right panel of Fig 1 we show the cumulative number of IC patient-days in excess as a function of time.The value reached at the final time, $t = T$, is our second QoI.

### Probability distributions of the QoIs

As first step of our UQ analysis, we construct the empirical cumulative distribution functions (cdfs) of our QoIs. For any threshold value $q^*$, the cdf gives the probability that the QoI remains below (or at) that threshold, i.e. it gives $\mathbb{P}(\text{QoI} \leq q^*)$ for the adopted strategy (and given the specific input parameter ditributions). In Fig 2 we report the resulting cdfs for the four selected strategies both with and without uncertainties in non-policy-related parameters (in the latter case, we fix these parameters at the mean values listed in Table 1, except for `shape_exposed_time` which is set to 20, see also S1 Table, and for the seed which is never fixed). We further display the 95%-confidence interval of the empirical cdfs using the Dvoretzky-Kiefer-Wolfowitz inequality [29, 30], which gives an indication on how reliable the empirical cdfs are.

We observe the important fact that, with the given distributions of the input parameters, none of the analyzed strategies is robust. The probability that the number of prevalent patients in intensive care is larger than the IC capacity is rather high and only Contact Tracing gets close to a probability of 50%. This shows that the assumed input distributions for Flattening the Curve, Intermittent Lockdown and Phased Opening correspond to interventions that are not sufficiently restrictive to stay below the threshold, as far as the model is concerned.

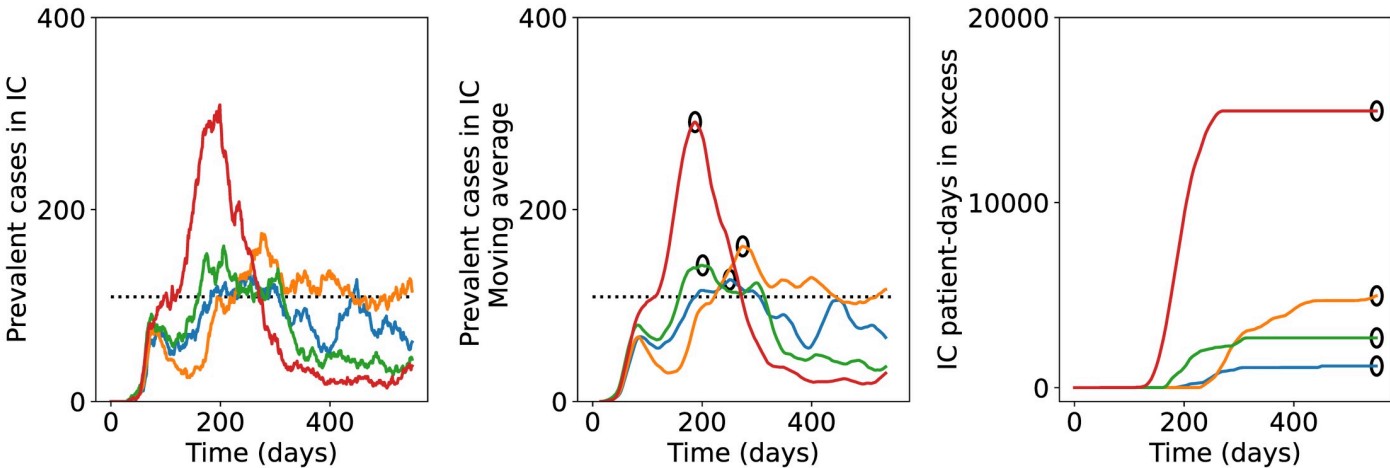

**Fig 1. Examples of model outcomes for four independent simulations of the Phased Opening strategy with different policy-related parameters.** Left: time series of the prevalent cases in intensive care. Middle: the resulting moving average of the number of IC prevalent cases. Right: the corresponding time series of the total number of IC patient-days in excess. All values on the vertical axis are per million capita. Black circles in the middle and right panel indicate the QoIs for the different simulations.

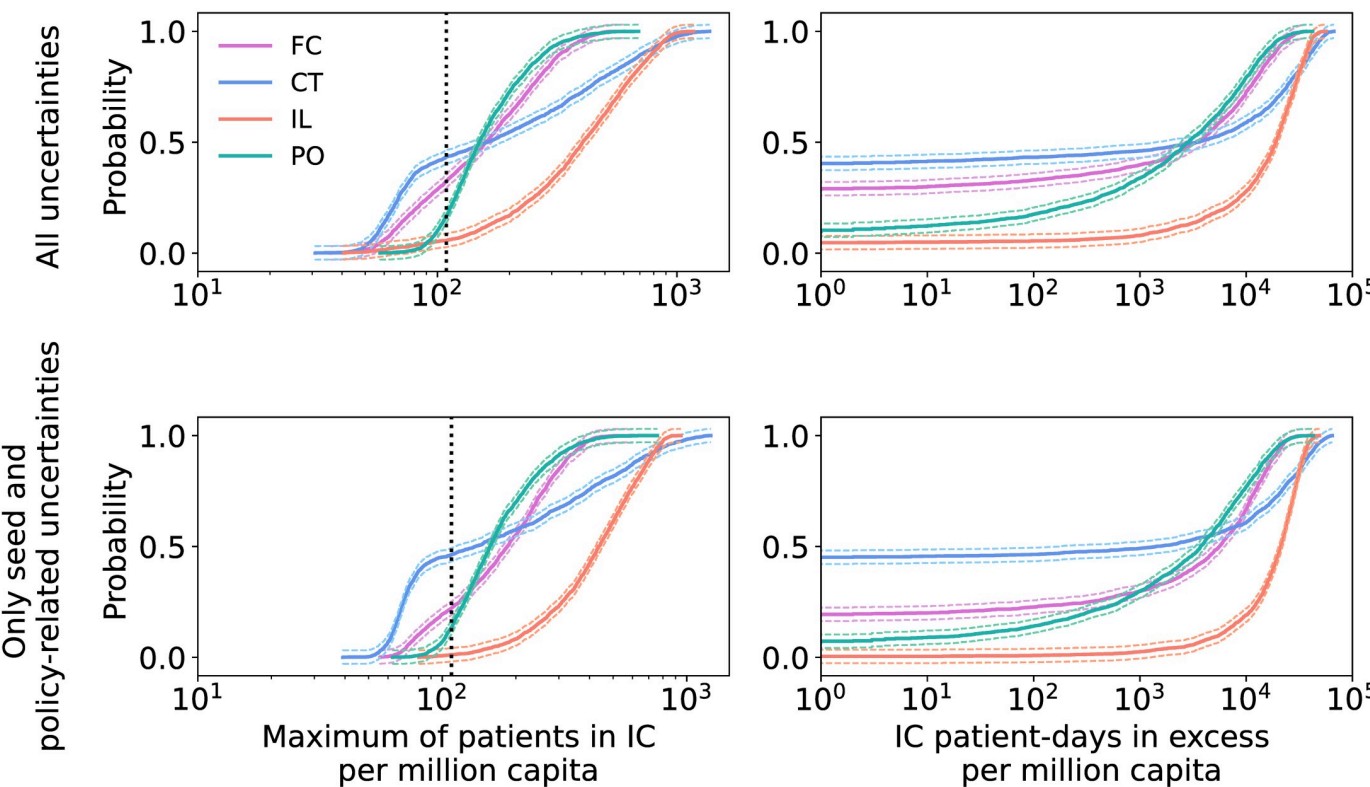

**Fig 2. Empirical cdfs for Flattening the Curve, Contact Tracing, Intermittent Lockdown and Phased Opening.** Top row: results with policy-related and non-policy-related uncertainties. Bottom row: outcomes with uncertainties only in the policy-related parameters and in the random seed. The vertical dotted black line indicates the maximum IC capacity, while the thinner colored lines denote the 95%-CI given by the Dvoretzky-Kiefer-Wolfowitz inequality. Note that the cdfs for the second QoI (right column) do not start from zero probability because the distributions have a non-zero probability that the number of IC patient-days in excess is zero.

We note that the cdfs of the first QoI for CT and IL increase more gradually compared to FC and PO (implying that the variance of the first QoI is larger for CT and IL than it is for FC and PO). Furthermore, it can be seen that the shape of the cdfs is only weakly affected by non-policy-related parameters. Therefore, when searching for the parameters responsible for most of the output variability, i.e. for the sensitivity analysis, these parameters might be kept fixed to reduce the computational burden. They should be included however when determining the minimum level of e.g. intervention or uptake, required for the QoIs to stay below their threshold with 95% probability.

## Sensitivity analysis

In Fig 3 we report, for the first QoI, the 95%-confidence interval of the first order Sobol indices for the policy-related parameters and for the seed. This information allows us to identify which factors of the strategies most affect the model output, such that measures targeting these factors can be adopted in order to make the strategy robust.

From the width of the intervals, it can be seen that the number of MC samples ($M = 2000$) is too small to estimate the Sobol indices very accurately. Nonetheless a qualitative ranking of the parameters with respect to the generated output variance can be made, based on the results in Fig 3. It is therefore possible to distinguish `intervention_effect`, `trace_rate_I`, `uptake` and `pl_intervention_effect_hi` as the parameters responsible for most of

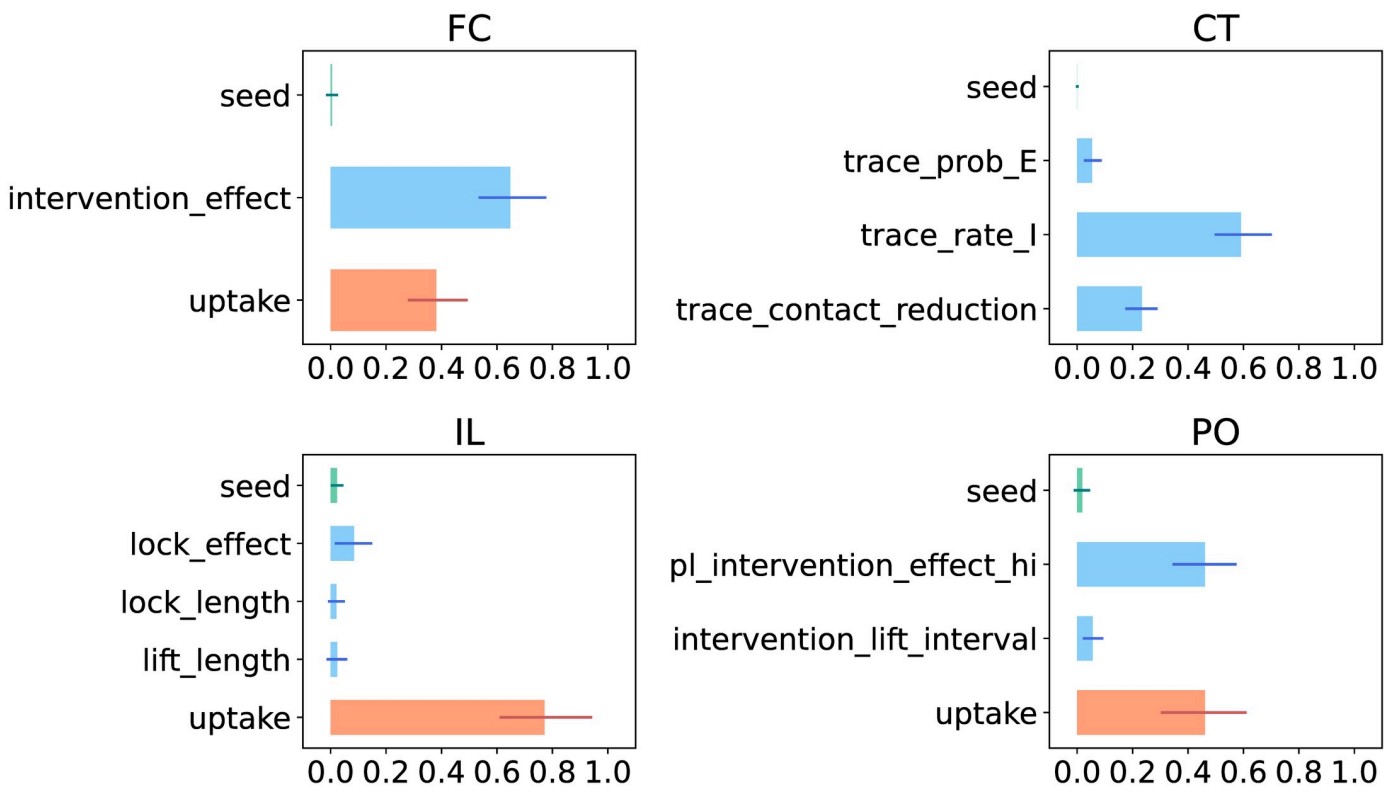

**Fig 3. First order Sobol indices of the first QoI (the maximum number of patients in IC).** The length of the bars indicate the mean values, while the thinner lines display the 95% confidence interval. We color in orange the `uptake` parameter, and in green `seed`.

the output uncertainty, and should thus be targeted (e.g. through policy measures) in order to obtain more desirable outcomes. For more accurate estimates of the Sobol indices, *M* must be increased substantially, thereby greatly increasing also the computational cost. We refrain from doing so in this study and give more comments on the associated computational cost in the Discussion.

The uptake by the population of the interventions plays a crucial role whenever this parameter is part of the strategy. In these strategies (i.e. FC, IL, PO), the `uptake` parameter is responsible for at least 30% of the QoI variance and, in case of Intermittent Lockdown it is responsible for about 70% of the variance. The `intervention_effect`, `lock_effect` and `pl_intervention_effect_hi` parameters are also important, although they are not always the main driving factor. For Flattening the Curve its Sobol index is higher than that of `uptake`, whereas for Intermittent Lockdown `uptake` has a much higher Sobol index (indicating that it is more important) than `lock_effect`. The population uptake and the effect of the lockdown seem to be of similar importance for the Phased Opening strategy.

In case of Contact Tracing circa 50% of the QoI variance is determined by the delay between becoming infectious and being identified as such (if at all), i.e. the inverse of the rate per day at which infected individuals are being traced (parameter `trace_rate_I`). After infected agents have been identified, the reduction of their average contact rate is also important.

The probability of tracing exposed individuals (CT strategy) and the interval between subsequent lifts (PO strategy) give only a small contribution to the model output variability. The intrinsic stochasticity of the model instead does not induce much variability in the model

output as the 95%-CI of its Sobol index always includes 0 and does not take values above 5%. Similar conclusions hold for the lengths of the lockdowns and subsequent lift periods in the IL strategy.

The total Sobol indices give qualitatively the same outcomes but highlight higher order interactions involving `lock_effect` and `trace_prob_E` (see S3 Fig). The differences between the first order and the total Sobol indices suggest that the most important second-order interactions appear to be those between `lock_effect` and `uptake` in the IL strategy, and between `trace_prob_E` and `trace_rate_I` in CT. Assessing these interaction effects in detail is computationally very expensive, therefore we do not carry out this assessment here. However, the scatter plots for the safe operating space give some idea of the interaction effects (see next section).

The Sobol indices for the second QoI show qualitatively similar results, see S4 and S5 Figs.

## Safe operating space

Given the knowledge on the main driving factors of each strategy, we want to know which combinations of values for these input parameters result in, for instance, a number of prevalent cases in IC (first QoI) below the IC capacity. This information can be used to devise policy measures that would effectively move the corresponding input distributions towards an area in the parameter space where the strategy is robust, i.e. towards the safe operating space. The information can be obtained by means of a scatter plot of the QoIs as functions of the values of the two or three input parameters with the highest Sobol indices.

In case of Flattening the Curve we visualize the two selected QoIs as functions of the most important parameters according to the Sobol indices, i.e. `intervention_effect` and `uptake`, see Fig 4. The first QoI stays below the IC capacity for high population uptake and strong intervention (recall that stronger intervention is associated with lower values of

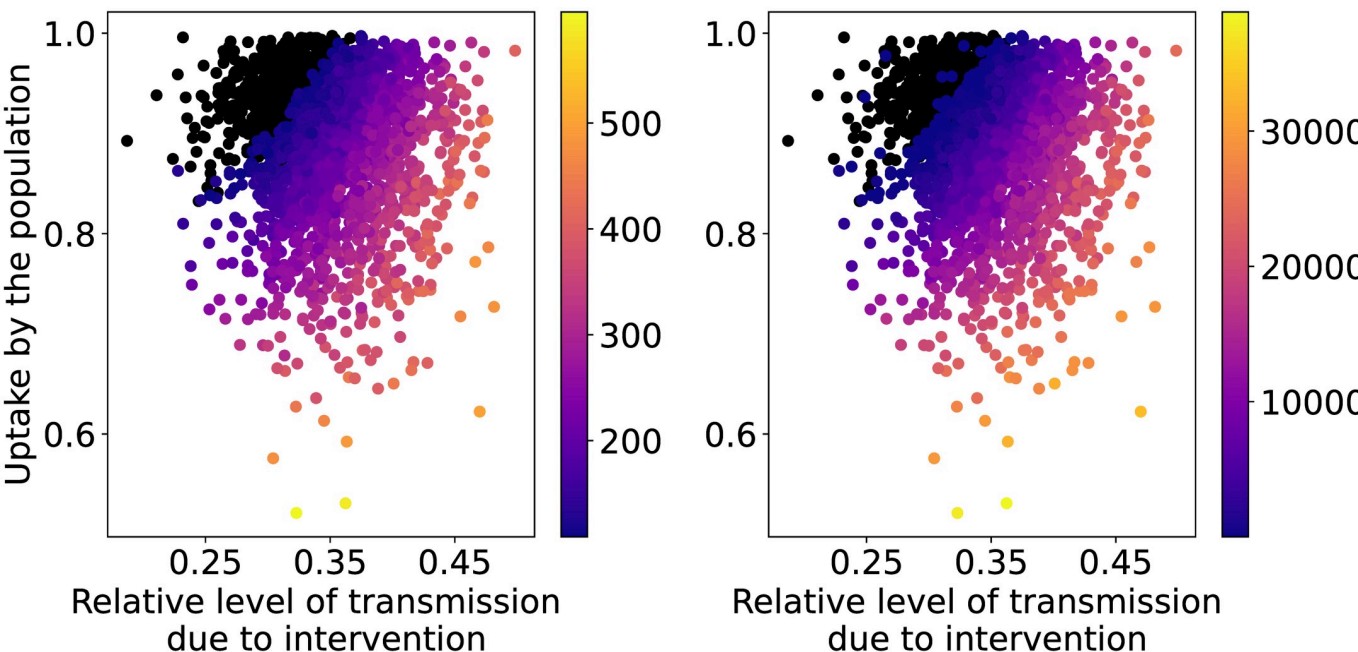

**Fig 4. Heat map of the maximum values of patients in IC (left) and the total amount of IC patient-days in excess (right) per million capita as functions of the input parameters for the FC strategy with seed and policy-related uncertainties.** The black dots show the simulations whose QoI value is below or equal to the IC capacity or there are no IC patient-days in excess.

`intervention_effect`). If `uptake` is less than circa 85% or if `intervention_effect` is higher than 35%, an exceedance of the IC capacity is to be expected in the model.

The two main drivers of the strategy Contact Tracing are `trace_rate_I` and `trace_contact_reduction`, while `trace_prob_E` has higher relevance when considered in combination with the other parameters. Therefore, we ignore the seed parameter as the variance it induces is small (the 95%-CI of its Sobol index being very tight around 0) and split our data into quartiles (very low, low, high and very high) of `trace_prob_E`. The scatter plots of the resulting sub-sets of data of the first QoI are provided in Fig 5. The same analysis for the second QoI reveals the same qualitative results and is therefore not reported here.

At low values of `trace_rate_I` (i.e. in case of a long delay in the identification process), the first QoI is predicted to be above the threshold independently from `trace_contact_reduction`. This is slightly mitigated in the plots corresponding to the high and very high quartiles of `trace_prob_E`, revealing the important interaction between these two parameters captured by the total Sobol index. It is therefore paramount to have an efficient tracing system for this strategy to be effective.

Out of the five considered parameters of the strategy Intermittent Lockdown, only two are important according to the estimated Sobol indices. These are `uptake` and `lock_effect`. The scatter plots of the QoIs as functions of these two parameters (see Fig 6) confirm that high levels of uptake are necessary when adopting this strategy. They also show that, as the level of transmission allowed by the lockdown increases (i.e. higher values of `lock_effect`), high levels of uptake are not enough to keep the first QoI below its threshold. Very little variability is induced by the other parameters.

Similar to Contact Tracing, the Phased Opening strategy has two parameters responsible for most of the output variance: `uptake` and `pl_intervention_effect_hi`. The interval between consecutive phases, i.e. `intervention_lift_interval`, has a small Sobol index, whereas the Sobol index of the random seed is very close to zero. We therefore apply a similar approach: we ignore the random seed and divide the data into quartiles of `intervention_lift_interval`. We show the scatter plots of the resulting sub-sets of the first QoI in Fig 7. The same analysis for the other QoI reveals the same qualitative insights and is therefore not reported here.

There is a clear gradient visible in Fig 7 with lowest QoI values obtained for high values of `uptake` and for low values of `pl_intervention_effect_hi` (corresponding to more restrictive lockdowns). Higher QoI values are obtained when either the population uptake or the rigor of the lockdown decreases. These two factors alone though are not enough to ensure that the IC capacity is not exceeded. In fact all of our simulations in the quartile corresponding to very low values of `intervention_lift_interval` go beyond the IC capacity (top left panel in Fig 7). By increasing `intervention_lift_interval`, more simulations stay below the IC capacity threshold.

## Discussion

The aim of this work was to perform a model-based quantitative analysis of the uncertainties and sensitivities of a number of selected exit strategies for the COVID-19 epidemic, thereby showing how such an analysis can be carried out and what decision-relevant insights can be obtained from it. We discussed how the analysis can be approached using methods and concepts from the field of Uncertainty Quantification and Sensitivity Analysis. UQ and SA make it possible to give probabilistic answers to policy questions, such as the probability that the number of prevalent cases in intensive care will not exceed the IC capacity. This helps

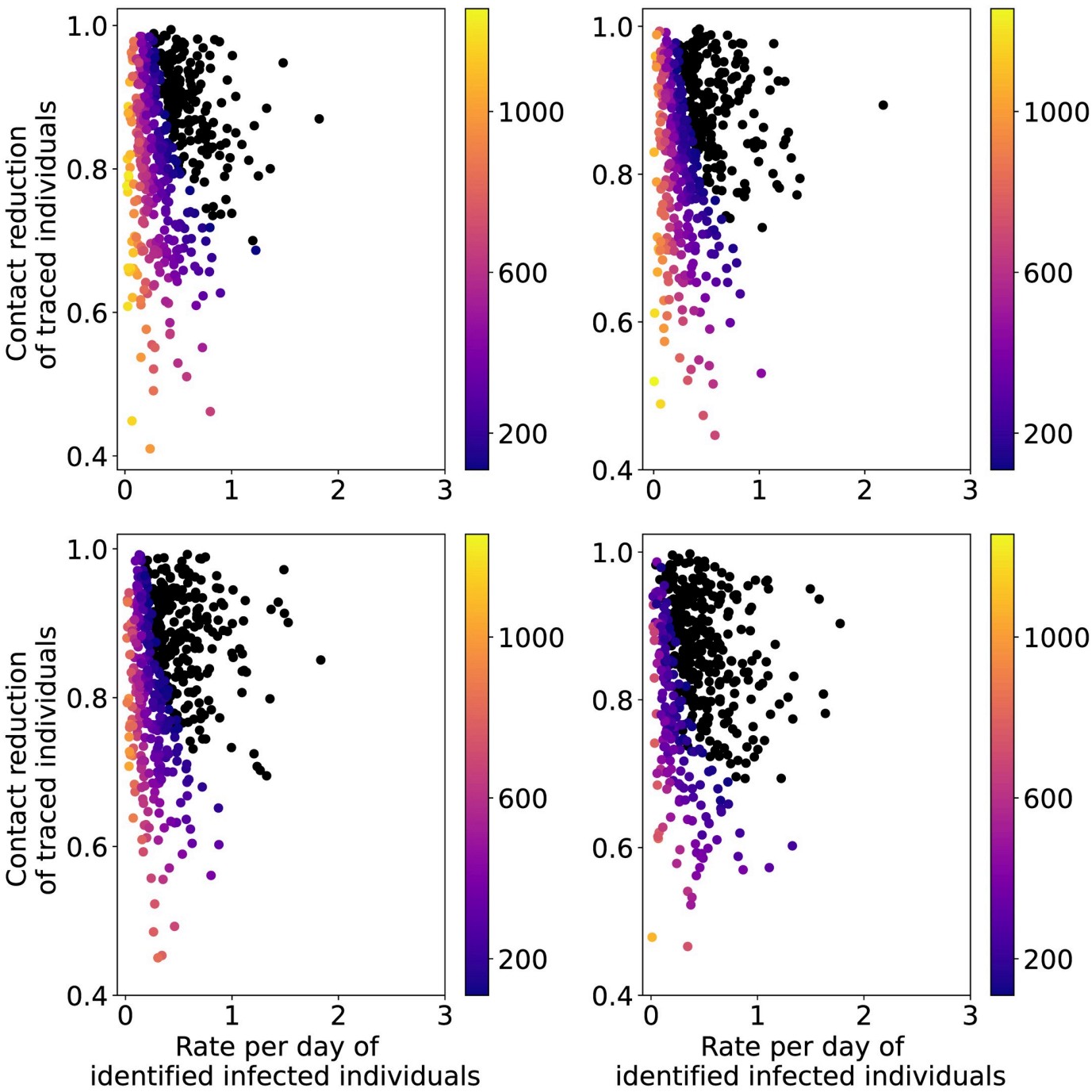

**Fig 5. Heat map of the maximum values of patients in IC per million capita as a function of the input parameters for the CT strategy with seed and policy-related uncertainties.** The black dots show the simulations whose maximum value is below or equal to the IC capacity. The plots correspond to quartiles of `trace_prob_E` (top left: very low; top right: low; bottom left: high; bottom right: very high).

policy makers to gauge the uncertainties in the model predictions and to take more informed decisions.

As concrete examples of decision-relevant insights that UQ and SA can deliver, we mention two observations taken from our numerical results. Fig 2 shows that from the four considered

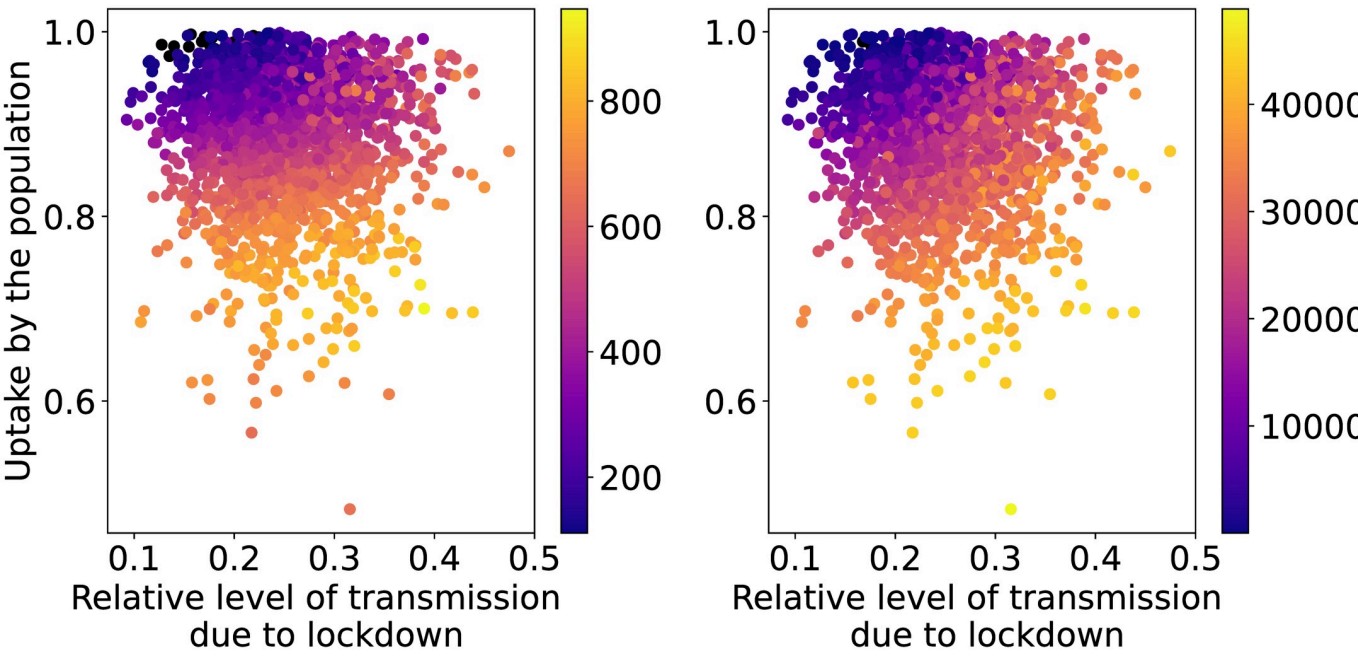

**Fig 6. Heat map of the maximum values of patients in IC (left) and the total amount of IC patient-days in excess (right) per million capita as functions of the input parameters for the IL strategy with seed and policy-related uncertainties.** The black dots show the simulations whose maximum value is below or equal to the IC capacity or there are no IC patient-days in excess.

strategies, CT has the highest probability that the peak number of IC patients remains below the maximum IC capacity of 109 beds per million capita. However, Fig 2 also shows that CT has a higher probability that the peak number reaches very high values (> 400) than the PO and FC strategies. Furthermore, the global SA results in Fig 3 demonstrate that for CT, the duration of the tracing process (parameter `trace_rate_I`) is more influential than the probability that an exposed individual gets traced (parameter `trace_prob_E`).

We demonstrated computational techniques that can be employed to assess the uncertainties and identify the sensitivities of each strategy. In particular, we examined the empirical cumulative distribution function of two quantities of interest obtained from the model output: the maximum number of prevalent cases in IC and the total amount of IC patient-days in excess of IC bed capacity. We also identified the input parameters responsible for most of the output uncertainty (namely `intervention_effect`, `trace_rate_I`, `uptake` and `pl_intervention_effect_hi`) via the computation of the Sobol indices of variance. Lastly we explored the shape of the safe operating space in the parameter space of the different strategies by means of scatter plots.

Given the probability distributions that we chose for the uncertain model parameters, the Contact Tracing strategy appears to have the most potential to avoid exceeding IC capacity (see Fig 2), but it requires high tracing capacity. Furthermore, the output uncertainty is highly driven by the uncertainty in the population uptake of, and adherence to interventions (see Fig 3), with the number of IC patients increasing as the value of `uptake` decreases and vice versa. The effect of interventions and the tracing capability also play a crucial role, whereas the intrinsic stochasticity of the model always has a low Sobol index so it gives only a minor contribution to the output uncertainty.

None of the strategies analyzed here satisfied our criterion of robustness with the given input distributions. This means that the probability that the number of prevalent patients in

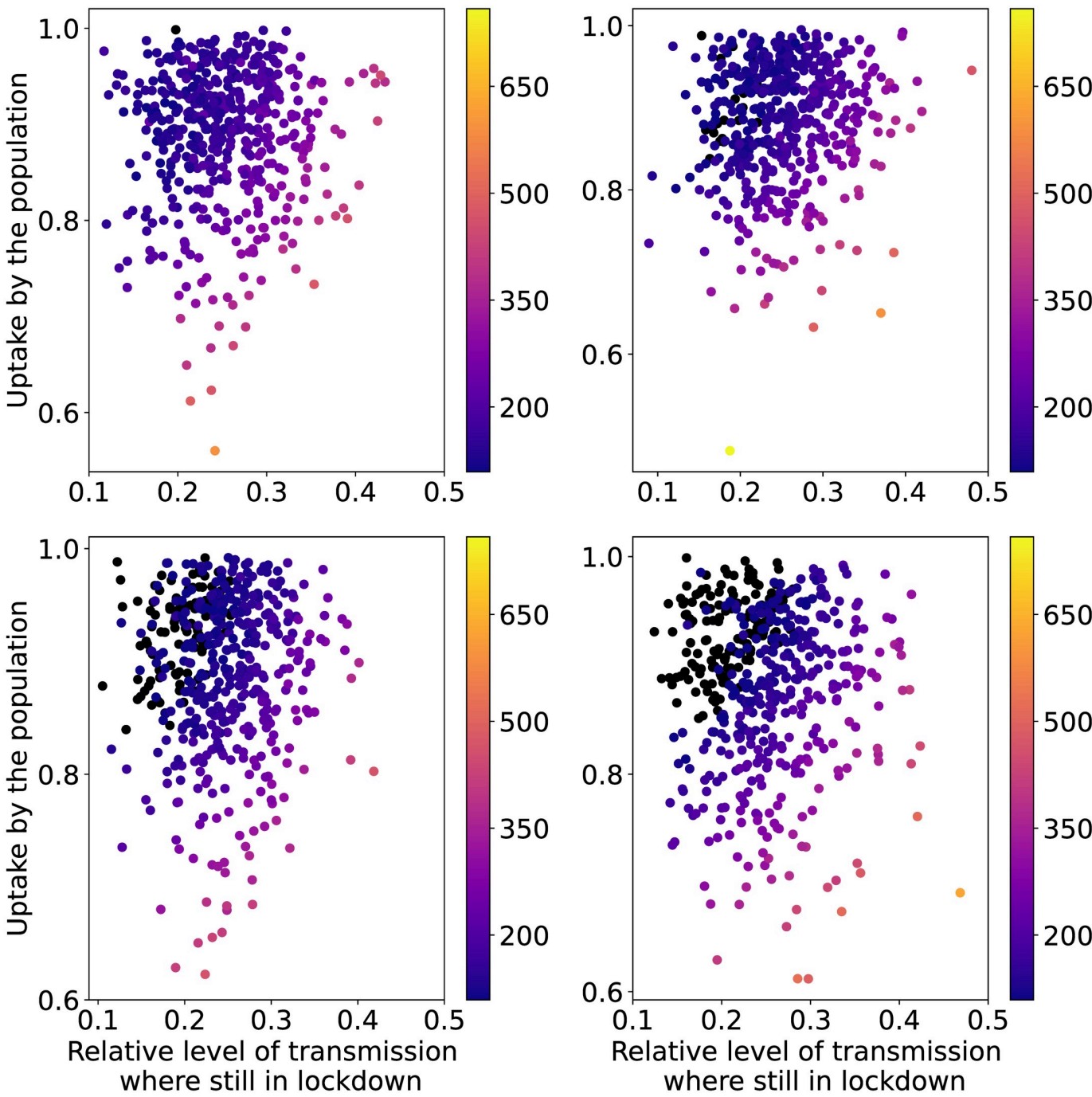

**Fig 7. Heat map of the maximum values of patients in IC per million capita as a function of the input parameters for the PO strategy with seed and policy-related uncertainties.** The black dots show the simulations whose maximum value is below or equal to the IC capacity. The plots correspond to quartiles (top left: very low; top right: low; bottom left: high; bottom right: very high) of `intervention_lift_interval`.

IC is larger than the IC capacity is rather high for all four exit strategies. To achieve more satisfactory performance (as far as the model and the chosen QoIs are concerned), parameter distributions corresponding to stricter interventions are needed. This would require that the effects of policy measures can "push" the bulk of the parameter distributions to more favorable

values compared to the distributions used in our analysis here. In particular the insights about the safe operating space can be useful to determine how the parameters distributions might need to be modified in order to obtain more desirable outcomes, e.g. towards stricter interventions in case of Flattening the Curve or towards longer intervals between consecutive phases for Phased Opening.

The analysis presented here can be extended to a broader set of models and diseases. Since we used non-intrusive methods, the same type of analysis can be applied to a different transmission model for COVID-19 or to a computational model for a different epidemic. Furthermore, the set of methods used for analysis can be enlarged in several ways. Many UQ and SA techniques are available tackling different aspects of the problem according to the type of information that one would like to obtain; see, e.g., [8, 31, 32]. For instance, in cases where the input parameters are not mutually independent, one can perform SA by computing the Kucherenko indices [33] instead of the Sobol indices. Another example is the computation of the partial rank correlation coefficients to obtain an estimate of how monotonous the relationship is between the input parameters and the QoIs [34]. Also, Bayesian inference can be used to estimate the distributions of (some of) the input parameters from data, or update these distributions as more data become available [7, 8] in a similar fashion as Taghizadeh et al. [35]. Finally, with Bayesian model averaging [36, 37] one can address model-form uncertainties (sometimes referred to as model structure uncertainties), and therefore compare the outcome of different models for the same epidemic.

A similar model-based analysis of uncertainties has been performed by Davies et al. [38], who used multiple realizations of their stochastic model in combination with variations in the basic reproduction number. Yet our analysis comprehends a larger set of uncertainties and is mathematically more rigorous as it is based on the theory and concepts of uncertainty quantification and sensitivity analysis. UQ techniques are not often applied to epidemiological models; exceptions are the work of Gilbert et al. [39] (who also discuss the relevance of uncertainty analysis to guide public health interventions) and the recent studies by Taghizadeh et al. [35] and by Edeling et al. [40]. However the analyses presented there have a different scope from our study: the analyzed model and the UQ methods differ, moreover our study encompasses a broader set of strategies and addresses extensively the intrinsic model uncertainty.

It is important to realise that UQ and SA results are conditioned on the choices of parameter distributions and should therefore be interpreted with caution. As such, our results should not be interpreted as a definitive formal ranking of the analyzed exit strategies, as these strategies might show better or worse performance when considering different distributions. We aimed at picking plausible distributions, however we do not claim a homogeneous level of rigor among the choices that were made. On a related note, we assumed in this study that once the strategy is started it is not changed, implying that input parameters (or their distributions) do not change over time. Furthermore, we limited the scope of our analysis to a small number of key parameters for which we assumed specific probability distributions. However, the computational model has additional parameters (relating both to policies and to other aspects), which were kept fixed here but could be added to the set of uncertain parameters in a more comprehensive UQ analysis.

A concrete example of these additional parameters is formed by the parameters specifying the geographical stratification. On one hand the geographical stratification makes the model more realistic and allows for heterogeneity in the population and the evaluation of regional measures; on the other hand it adds parameters to the model, making it more uncertain. Thus, the importance of the geographical stratification for policy making should be assessed case by case. In our case it was important because one of the considered strategies (Phased Opening) includes regional measures. If regional interventions are not under consideration, the benefits of the geographical stratification may not outweigh the additional uncertainty implied by the

(uncertain) stratification parameters. This relates to the more general issue of model selection and of balancing model complexity against model uncertainty and risk of overfitting, a well-known challenge in mathematical and statistical modeling.

For the policy-related parameters, the chosen distributions correspond to the (assumed) effects of policy measures in the real world. The feasibility of implementing such measures is a different matter, beyond the realm of mathematical and computational modeling and therefore not considered here, but important nonetheless. As a concrete example, Fig 5 shows that the delay in the identification of infected individuals must be short for the number of IC patients to stay below IC capacity. This result agrees well with Hellewell et al. [41], who found that when the delay between symptoms onset and isolation increases, the probability to keep the spread of the virus under control decreases. However, achieving such high levels of effectiveness may prove challenging in reality, given the increasing level of population fatigue with regard to policy adherence.

We conclude with some remarks about scaling up the analysis performed here to more extensive assessments and to more complex models (with higher computational cost). Scaling up will enable fast, frequent and comprehensive analysis of uncertainties and sensitivities in epidemiological models. Executing such analysis in a timely fashion is essential for it to be useful for policy makers. In this study we limited the analysis to a handful of uncertain parameters (less than 10), keeping all other parameters fixed. A more comprehensive study, on a larger set of parameters, will be computationally more demanding as it typically requires more model runs. While the execution of a single simulation of the virsim model takes 1–2 minutes on a laptop, more complex models have higher computational costs. Furthermore, (tens of) thousands simulations are required for thorough analysis of model uncertainties and sensitivities. Thus, access to sufficient computational resources is important to scale up.

For the analysis reported here, we had access to a supercomputer of the Poznan Supercomputing and Networking Center. A single campaign with circa 10000 runs for the SA of an exit strategy took several hours in total (including the time needed for the job submission to the supercomputer, parallel execution of the model runs on a single node with 28 cores, and retrieval of the results). If quantification of uncertainties is to be performed frequently and rapidly (e.g. in an "operational" setting with a daily or weekly cycle of producing forecasts with quantified uncertainties, or for weekly re-evaluation of a multitude of policy options), a dedicated computational infrastructure is recommendable to have uninterrupted access and to avoid long queuing times for compute jobs.

Besides access to computational resources, software suitable for efficient UQ and SA is needed. The open source VECMA toolkit used in this study is developed for use on high-performance computing platforms. Last but not least, a dedicated team with combined expertise (UQ and SA; epidemiology and computational modeling; high-performance computing and software) will be central to successful upscaling and thereby to support policy making with timely information about uncertainties and sensitivities of model results.

## Conclusions

In this study we analyzed the uncertainties and sensitivities of an agent-based transmission model for the COVID-19 epidemic under four different exit strategies. Our analysis showed that the uncertainties in the model simulation results for each considered exit strategy are substantial. They were found to be mostly generated by uncertainties in the parameters directly related to the strategy itself (such as implementation and uptake of the strategy) rather than uncertainties due to other factors (such as duration of infectiousness). With the parameter distributions that we choose, the Contact Tracing strategy was the most effective. Finally, because

we used non-intrusive methods, our analysis can easily be extended to other strategies as well as to other computational models and epidemics.

## Supporting information

**S1 Table. Overview of parameter values used in the analysis.** Overview of the values of the parameters in the SEIR model. In the last column we indicate the respective parameter in the computational model (or which computational parameter is affected).
(PDF)

**S1 Text. Technical model description of virsim.**
(PDF)

**S2 Text. Numerical implementation of the strategies.**
(PDF)

**S3 Text. Theory of Sobol indices.**
(PDF)

**S1 Fig. Distributions of the uncertain input parameters.** Distributions for the uncertain input parameters in case of Flattening the Curve (top left), Contact Tracing (top right), Intermittent Lockdown and Phased Opening (middle), and for the biology-related parameters (bottom).
(TIF)

**S2 Fig. Example of strategy outcome variability due to intrinsic stochasticity.** Example of strategy outcome variability due to intrinsic stochasticity of the virsim model. The shaded gray area denotes the interval between the $5^{th}$ and the $95^{th}$ percentiles out of 100 realizations of the FC strategy with same policy- and non-policy-related parameters but different `seed`. A few individual realizations are shown in colored lines.
(TIF)

**S3 Fig. Total Sobol indices of the first QoI.** Total Sobol indices of the first QoI (the maximum number of patients in IC). The length of the bars indicate the mean values, while the thinner lines display the 95% confidence interval. We color in orange the `uptake` parameter, and in green `seed`.
(TIF)

**S4 Fig. First order Sobol indices of the second QoI.** First order Sobol indices of the second QoI (the total number of IC patient-days in excess of IC bed capacity). The length of the bars indicate the mean values, while the thinner lines display the 95% confidence interval. We color in orange the `uptake` parameter, and in green `seed`.
(TIF)

**S5 Fig. Total Sobol indices of the second QoI.** Total Sobol indices of the second QoI (the total number of IC patient-days in excess of IC bed capacity). The length of the bars indicate the mean values, while the thinner lines display the 95% confidence interval. We color in orange the `uptake` parameter, and in green `seed`.
(TIF)

## Acknowledgments

The calculations were performed at the Poznan Supercomputing and Networking Center.

## Author Contributions

**Conceptualization:** Federica Gugole, Luc E. Coffeng, Wouter Edeling, Benjamin Sanderse, Daan Crommelin.

**Data curation:** Federica Gugole.

**Formal analysis:** Federica Gugole.

**Funding acquisition:** Luc E. Coffeng, Sake J. de Vlas, Daan Crommelin.

**Investigation:** Federica Gugole.

**Methodology:** Daan Crommelin.

**Project administration:** Daan Crommelin.

**Resources:** Daan Crommelin.

**Software:** Federica Gugole, Wouter Edeling.

**Supervision:** Daan Crommelin.

**Validation:** Federica Gugole, Benjamin Sanderse.

**Visualization:** Federica Gugole.

**Writing – original draft:** Federica Gugole, Luc E. Coffeng, Wouter Edeling, Benjamin Sanderse, Sake J. de Vlas, Daan Crommelin.

**Writing – review & editing:** Federica Gugole, Luc E. Coffeng, Wouter Edeling, Benjamin Sanderse, Sake J. de Vlas, Daan Crommelin.

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
