## [Decision Letter · Decision Letter 0]

10 May 2021

Dear Dr Gugole,

Thank you very much for submitting your manuscript "Uncertainty quantification and sensitivity analysis of COVID-19 exit strategies in an individual-based transmission model" for consideration at PLOS Computational Biology.

As with all papers reviewed by the journal, your manuscript was reviewed by members of the editorial board and by several independent reviewers. In light of the reviews (below this email), we would like to invite the resubmission of a significantly-revised version that takes into account the reviewers' comments.

The reviewers were in general enthusiastic about the topic and the quaility of writing. I too think that uncertainty analysis in modelling policy advise is important, but I think the manuscript currently does not discuss practical implementation sufficiently. It is said that the model is conceptual, and are an illustration of the UQ and SA approaches, but the results and especially discussion are pretty much focussed on the results of the COVID-19 control scenarios, and less about what the UQ and SA would contribute to the decision making process. I think that should be central: when and why use UQ, and when and why use SA, and how does this improve decisions about the right policy to choose? After all, the manuscript was submitted for the Methods section, and although COVID-19 itself is topical, the strategy comparison in this manuscript is not anymore, so it really should serve as an illustration.

We cannot make any decision about publication until we have seen the revised manuscript and your response to the reviewers' comments. Your revised manuscript is also likely to be sent to reviewers for further evaluation.

Sincerely,

Don Klinkenberg

Associate Editor

PLOS Computational Biology

Virginia Pitzer

Deputy Editor-in-Chief

PLOS Computational Biology

Reviewer's Responses to Questions

**Comments to the Authors:**

Reviewer #1: This is a very well-written paper and I read it with enthusiasm. I had a few suggestions:

1. The intermittent lockdown is assumed to go into effect periodically with a prespecified duration of lockdown and then a prespecified duration of opening. I am not sure if this is realistic. If possible, I'd suggest modeling this on/off decision as a function of some epidemic threshold; for example, lockdown when the number of COVID-19 patents in ICU passes a threshold T1 and reopen when the number of COVID-19 patients in ICU goes below a threshold T2. This is more consistent with the way decisions about physical distancing measures have been made and modelled in existing studies.

2. Please explain briefly why the 'random number seed' is also considered as parameter (line 346). I am assuming this is to ensure that the Sobel index of the random number seed is negligible since otherwise, it suggests a problem with the random number generator.

3. There are other approaches for sensitivity analysis and quantification of uncertainty such as the use of partial-rank correlation: https://www.ncbi.nlm.nih.gov/pmc/articles/PMC3730677/. I think the paper would benefit from a more detailed discussion of alternative methods for SA and QU.

Reviewer #2: This manuscript provides a well-done and much needed discussion of

public-health measures against the COVID-19 pandemic. It is extremely

timely and provides well-founded answers to the highly relevant

question how effective the various policy measures are. The numerical

results can be improved given more computational resources.

Major points that should be addressed in a revision are the following.

Regarding the computational model (page 3), it would be convenient for

future readers to show the (well-known) SEIR model equations and

especially how the geographical stratification works. This could also

be done in an appendix, but I believe it would be preferable to show

the basic equations and the geographical stratification at the

beginning to help the reader to understand how and where the various

parameters enter the system of equations. In this regard, it would

also be useful to move Table 1 up close to the equations so that the

reader can find this important information, i.e., the system of

equations and all parameters, in one place at the beginning.

On page 3, it is said that "this study is conceptual in nature".

Still, it does a very good job at trying to be realistic. A

discussion of what is still needed to move from a concept to a

realistic treatment would be beneficial in order to help the reader

understand any shortcomings if they exist.

On pages 6 ff., it is mentioned that Beta and Gamma distributions are

assumed for the parameters and some justification is given on page 6.

However, would it be possible to provide more specific justifications

on these pages where the parameters are discussed?

In Table 1, references (if available) to justify the choice of the

parameter values would be very welcome and add a lot to the

discussion.

On page 9, it is mentioned that the geographical stratification in the

model adds to the uncertainty in the model via additional parameters.

More unknown parameters mean more uncertainties. Therefore the

question arises if the additional geographical parameters are indeed

advantageous in the sense that they yield an improved model. Does the

improvement in the model justify the complications? These questions

should be discussed.

On page 12 it is mentioned that the number M of runs would need to be

increased substantially for more accurate estimates of the Sobol

indices. Therefore the computational cost (CPU hours) should be

discussed a bit more detailed. The discussion in the discussion

section comes a bit late and I think it would be good to move some

statements about the computational cost up.

On page 17, the references [7, 8] regarding the use of Bayesian

inference to estimate the distributions of the input parameters are

very general. Also reference [35] is cited in the context that UQ

techniques have (surprisingly) seldom been applied to epidemiological

models. An exception I could find is the following work, where such

Bayesian calculations were performed already a year ago; the authors

might want to consider citing it, also in the context of Table 1.

[Leila Taghizadeh, Ahmad Karimi, and Clemens Heitzinger. Uncertainty

quantification in epidemiological models for the COVID-19

pandemic. Computers in Biology and Medicine, 125(104011):1--11, 2020.]

In the discussion on page 18, the computational cost is approximately

given. More data would be very useful. How long does one simulation

take? Also, is the geographic stratification in the simulations worth

it? It has the drawback that it increases computational cost and adds

more (uncertain) parameters. Also it seems as if the geographical

part of the model was left constant in this work anyway.

Reviewer #3: Gugole et al. present a thorough uncertainty quantification and sensitivity analysis of an agent-based model of SARS-CoV-2 transmission. The model is based on the Netherlands, but the paper is really focused on describing these methods carefully and discussing the interpretation of their output, rather than attempting to make realistic projections. They consider four strategies for reopening, and find that, for each strategy, 2-3 key parameters largely determine whether ICU capacity will remain manageable, and that these key parameters are always policy-related parameters.

I very much enjoyed reading the paper. It is well written, explains the methods clearly, and would be a very useful read for an infectious disease modeler (or any modeler) who wishes to learn and employ UQ and SA in their work. I only have a few minor comments, detailed below:

Methods and model

• Line 73: you could refer to S1 Fig here, as the reader may wish to visualize the distributions of these parameters.

• What sampling method did you use to sample from the parameters distributions in the UQ?

• Line 208: write the date to which you reconstruct rather than ‘present time’

• Please give references for the distributions of the non-policy related parameter ranges you use, e.g. R0, infectiousness duration etc. If possible, it would be nice to see references supporting your choice of distributions for the policy related parameters, too. I think at the moment, the rather hand-wavy justification for these distributions may be the weakest part of the manuscript, although I appreciate that most of these parameters may not have much literature to support your choices.

• Line 337: Could you explain your choice of 1000 simulations for the UQ and the choice of M=2000 for the SA.

• On this subject, it’s not completely clear whether you do 1000 simulations per parameter set or 1000 overall? I’m assuming the former, but probably best to make this clear around line 338

• In Table 1, could you indicate in some way which type of parameter each is (i.e. policy-related or other), and if policy-related, which strategies it pertains to?

Results

• Could you show a supplementary figure of one reopening scenario, similar to Figure 1, but where each line uses the same parameter set? Perhaps with more than four lines – something like a spaghetti plot would work well. This would help the reader to get a sense of how stochastic the model is.

• Line 407: could you include the SA for the other QoI as a supplementary figure?

• Figure 3: Could you show the higher order terms on this plot? Judging from the x-axis they must be pretty small, but it would still be nice to see it.

• Line 443: Could you show the second order interaction terms in a supplementary table or figure (e.g. a heat-map)?

Discussion

• Line 512: I’m not sure most effective is the correct term here, because as you say, it depends on the parameters. Perhaps it would be better to say has the most potential to avoid exceeding ICU capacity, or something like that?

**Have the authors made all data and (if applicable) computational code underlying the findings in their manuscript fully available?**

Reviewer #1: Yes

Reviewer #2: Yes

Reviewer #3: Yes

PLOS authors have the option to publish the peer review history of their article (what does this mean?). If published, this will include your full peer review and any attached files.

Reviewer #1: No

Reviewer #2: No

Reviewer #3: No
---

## [Decision Letter · Decision Letter 1]

17 Aug 2021

Dear Dr Gugole,

We are pleased to inform you that your manuscript 'Uncertainty quantification and sensitivity analysis of COVID-19 exit strategies in an individual-based transmission model' has been provisionally accepted for publication in PLOS Computational Biology.

Best regards,

Don Klinkenberg

Associate Editor

PLOS Computational Biology

Virginia Pitzer

Deputy Editor-in-Chief

PLOS Computational Biology

Reviewer's Responses to Questions

**Comments to the Authors:**

Reviewer #1: The authors have addressed all my comments and concerns in this revision.

Reviewer #2: The points raised in my review were addressed.

**Have the authors made all data and (if applicable) computational code underlying the findings in their manuscript fully available?**

Reviewer #1: Yes

Reviewer #2: Yes

PLOS authors have the option to publish the peer review history of their article (what does this mean?). If published, this will include your full peer review and any attached files.

Reviewer #1: No

Reviewer #2: No

---

## [Editor Report · Acceptance letter]

14 Sep 2021

PCOMPBIOL-D-21-00514R1

Uncertainty quantification and sensitivity analysis of COVID-19 exit strategies in an individual-based transmission model

Dear Dr Gugole,

I am pleased to inform you that your manuscript has been formally accepted for publication in PLOS Computational Biology. Your manuscript is now with our production department and you will be notified of the publication date in due course.

With kind regards,

Andrea Szabo
